# Respiratory Muscle Interval Training Improves Exercise Capacity in Obese Adolescents during a 3-Week In-Hospital Multidisciplinary Body Weight Reduction Program

**DOI:** 10.3390/ijerph20010487

**Published:** 2022-12-28

**Authors:** Desy Salvadego, Gabriella Tringali, Roberta De Micheli, Alessandro Sartorio

**Affiliations:** Istituto Auxologico Italiano, Istituto di Ricovero e Cura a Carattere Scientifico (IRCCS), Experimental Laboratory for Auxo-Endocrinological Research, 28824 Piancavallo-Verbania, Italy

**Keywords:** respiratory training, exercise tolerance, adolescents with obesity

## Abstract

The purpose of this study was to determine whether a novel approach of interval training targeted to the respiratory muscles (RMIT; normocapnic hyperpnea with resistance) in addition to a multidisciplinary in-hospital body weight reduction program (BWRP) was able to improve the integrative response to exercise in young patients with obesity. Nine male patients (17.9 ± 4.9 (x ± SD) years; 113.8 ± 16.3 kg) underwent 12 sessions of RMIT and eight age-and sex-matched patients underwent 12 sessions of a sham protocol (CTRL) during the same 3-week BWRP. Before and after the interventions the patients performed an incremental and a heavy-intensity constant work-rate (CWR>GET) cycling exercise to voluntary exhaustion. Body mass decreased by ~4.0 kg after both RMIT (*p* = 0.0001) and CTRL (*p* = 0.0002). Peak pulmonary O_2_ uptake ($\dot{V}$O_2_) increased after RMIT (*p* = 0.02) and CTRL (*p* = 0.0007). During CWR>GET at ISO-time, $\dot{V}$O_2_ (*p* = 0.0007), pulmonary ventilation (*p* = 0.01), heart rate (*p* = 0.02), perceived respiratory discomfort (RPE_R_; *p* = 0.03) and leg effort (*p* = 0.0003) decreased after RMIT; only RPE_R_ (*p* = 0.03) decreased after CTRL. Time to exhaustion increased after RMIT (*p* = 0.0003) but not after CTRL. In young patients with obesity, RMIT inserted in a 3-week BWRP reduced the cardiorespiratory burden, the metabolic cost, the perceived effort, and improved exercise tolerance during heavy-intensity cycling.

## 1. Introduction 

Obesity has profound effects on the respiratory function, even in the absence of primary parenchymal lung or restrictive chest wall diseases. Respiratory dysfunctions observed in patients with obesity are characterized by impaired breathing mechanics, decreased respiratory system compliance, increased small airway resistance, and by alterations in both breathing pattern and respiratory drive [1,2,3,4]. Such dysfunctions are already evident in children and adolescents who do not exhibit complications induced by the disease [5,6,7,8]. Overall, these alterations can cause increased work of breathing and O_2_ cost of breathing [9,10] and increased metabolic requirements at rest and during exercise [11], and concur to determine exertional dyspnea and early respiratory muscle fatigue during exercise [12]. Moreover, these alterations are frequently associated with respiratory muscle weakness [13], which may exacerbate the cascade of negative effects on the integrative response to exercise and ultimately the capacity to accomplish exercise and daily-life physical activities. Hence, the importance to specifically address part of the preventive/therapeutic interventions to the respiratory muscles. To date, however, information on the pathophysiological effects of breathing exercises/respiratory muscle training in patients with obesity are still scanty, especially in young patients, and there is no reference guideline indicating which characteristics these interventions should have (modality, doses, timing) to be safely and efficiently recommended to the patients [14,15]. 

The effectiveness of an endurance training protocol for the respiratory muscles (respiratory muscle endurance training, RMET) in adolescents with obesity has been suggested by Frank et al. [16], who observed a reduced sensation of breathlessness during exercise and an increase in the distance covered during a 12-min time trial after 7 months of RMET, and subsequently by our studies [5,17,18], which found a significant reduction in the metabolic cost of cycling and uphill walking accompanied by lower perception of breath effort and lower respiratory muscle burden after 3 weeks of RMET associated with a multidisciplinary body weight reduction program. RMET consists of volitional normocapnic hyperpnea at high levels of ventilation continuously sustained for ~20–30 min [19]. Because of its characteristics of endurance training, an RMET program requires the participants to dedicate a significant amount of time and motivation, which may not fit the consensus of the individuals. To overcome these complications, a high-intensity interval training protocol has been recently developed for the respiratory muscles by combining normocapnic hyperpnea at high levels of ventilation with high inspiratory and expiratory pressures [20]. This novel protocol was able to improve significantly the respiratory muscle function of healthy young individuals with significantly lower training volumes than the classic RMET protocol [21], thus laying the groundwork for its effectiveness in patient populations affected by respiratory dysfunctions and exercise intolerance.

On the basis of these premises, the aim of this study was to assess whether, in a group of young patients with obesity, an adapted 12-session program of interval training targeted to the respiratory muscles, added to a 3-week in-hospital multidisciplinary body weight reduction program, was effective to improve exercise capacity and to alleviate the cardiorespiratory burden, the O_2_ demand and the perception of effort during heavy-intensity cycling. If so, the program of respiratory muscle interval training (RMIT) could be usefully applied and prescribed as an additional intervention in the preventive/therapeutic programs for the treatment of obesity/overweight, both in hospital and at home.

## 2. Materials and methods 

### 2.1. Population 

Twenty male young patients with obesity (age: 17.2 ± 5.0 years), hospitalized at the Divisions of Auxology and Metabolic Diseases, Istituto Auxologico Italiano, IRCCS, Piancavallo-Verbania, Italy for a multidisciplinary 3-week body weight reduction program (BWRP), were admitted to the study and randomly assigned to two different rehabilitation programs. The randomization sequence was computer-generated and patients were blinded to treatment allocation. Three patients did not complete the training schedule and/or the exercise tests after the interventions because physical injuries/sickness occurred during the hospitalization, thus were not included in the analyses. The first group was exposed to the intervention (RMIT, *n* = 9; 17.9 ± 4.9 years; body mass 113.8 ± 16.3 kg) and carried out a specific program of respiratory muscle interval training. The second group (CTRL, *n* = 8; 17.0 ± 5.7 years; body mass 109.3 ± 8.9 kg) carried out a sham protocol of respiratory training as comparison. Fat-free mass (FFM) and fat mass (FM) were determined by a multifrequency impedance meter (Human IM touch^®^, DS-Medica, Milan, Italy) in the fasting state [22,23,24]. The coefficient of variation within groups for FFM and FM was ~5%, while the intraclass correlation coefficient was 0.8%. 

The inclusion criteria were: (1) BMI > 97th percentile for age and sex, using the Italian growth charts [25] for patients aged < 18 years or BMI > 35 for patients > 18 years; (2) no involvement in structured physical activity programs (regular activity more than 120 min week^−1^) during the 8 months preceding the study; (3) absence of signs or symptoms of diabetes or of any major cardiovascular, respiratory or orthopaedic disease. 

After being fully advised about the purposes and testing procedures of the investigation, the recruited patients gave their assent to participate. Participants or participants’ parents/legal caregivers provided signed informed consent statements. All procedures were in accordance with the recommendations set forth in the Helsinki Declaration (2000) and with the Additional Protocol to the European Convention of Human Rights and Medicine concerning Biomedical Research (2005), and were approved by the ethics committee of Istituto Auxologico Italiano, Milan, Italy (ref. No. 2020_02_18_09; date of approval 18 February 2020). 

### 2.2. Body Weight Reduction Program

The participants were admitted as in-patients for a 3-week in-hospital multidisciplinary BWRP [5], entailing the following: (a) Moderate energy restriction, with a personalized diet entailing an energy intake ~500 kcal lower than the measured resting energy expenditure. Diet composition was formulated according to the Italian recommended daily allowances (Società Italiana di Nutrizione Umana); daily caloric intake was established by a dietician, who carefully supervised food consumption and the compliance to the diet. In terms of macronutrients, the diet contained approximately 21% proteins, 53% carbohydrates and 26% lipids, the daily estimated water content was 1000 mL, whereas the estimated salt content was 1560 mg Na+, 3600 mg K+ and 900 mg Ca2+. Extra water intake of at least 2000 mL/day was encouraged. (b) Aerobic exercise training, carried out under the guidance of a therapist, who completed a daily diary with the type, amount and intensity of the exercise performed by the subjects. All the participants followed a whole-body exercise training program including two 30-min sessions/day of cycling, treadmill walking, and stationary rowing, preceded and followed by ~10-min stretching, for 5 days/week, with heart rate (HR) monitoring and medical supervision. The initial intensity of exercise was set at ~60% of HRpeak determined during the incremental exercise test before the intervention, and was progressively increased reaching ~80% at the end of the exercise program. While RMIT (see below) was performed in the morning, whole body exercise training was administered in the afternoon. (c) Psychological and nutritional counselling, which included individual clinical sessions (1–2/week) and psycho-educational working groups focused on motivational aspects and on the control of self-behaviours for health and stress management.

### 2.3. Respiratory Muscle Interval Training 

The respiratory muscle training was characterized by an interval protocol of intensive breathing (RMIT). This type of training was performed by using a novel device (P100, Idiag, Fehraltorf, Switzerland) allowing for increases in both the breathing frequency and the mouth resistance. The specific properties of the device allow for personalized respiratory training mixing strength and endurance through intensive inspirations and expirations, without hypocapnia [20,21]. 

The patients of the RMIT group completed 12 interval training sessions over 3 weeks (4 days per week). Each session consisted of 6 bouts of 2-min vigorous breathing followed by 1 min of spontaneous breathing. During intensive breathing, the subjects breathed through the device characterized by an adaptive resistance set at 40% of the individual maximal inspiratory and expiratory pressure, determined with the same device immediately before the training program. The selected resistance allowed the patients to sustain a breathing frequency of 25 breaths per minute with a tidal volume corresponding to 60% of the forced vital capacity. 

The patients of the CTRL group completed a 12-session program of sham respiratory training. The characteristics of the training were similar but the patients used a modified volumetric exerciser (Covidien, Istanbul, Turkey) with no resistance to ventilation, and maintained a breathing frequency of 15 breaths per minute during the working intervals.

All training sessions were performed under the supervision of the personnel of the Experimental Laboratory for Auxo-Endocrinological Research who were specifically trained in using the device and conducting the training. 

The perception of respiratory effort, heart rate, arterial blood O_2_ saturation (by pulse oximetry), and systolic and diastolic blood pressures were regularly controlled throughout the training sessions. Data obtained during the first and the last sessions are described in Table 1. No symptoms of lightheadedness or malaise were described by any patient during or after the sessions. 

### 2.4. Spirometry

Before and after the interventions the patients performed standard spirometric tests (forced vital capacity, FVC; forced expiratory volume in 1 s, FEV_1_; FEV_1_/FVC; forced expiratory flow between 25% and 75% of FVC, FEF_25–75%_; peak expiratory flow, PEF) by utilizing the same metabolic cart (MedGraphics CPX/D, Medical Graphics Corp., St Paul, MN, USA). Pulmonary function testing was performed according to the guidelines of the American Thoracic Society [26]. Patients were carefully instructed about the spirometric manoeuvres and were allowed time to practice with the technique before performing the tests. Predicted values were based on Hankinson et al. [27].

### 2.5. Exercise Testing 

A series of exercise tests was conducted under medical supervision within 3 days before and after the interventions (see 18 for details). During the exercise tests, the patients were continuously monitored with 12-lead electrocardiography (ECG). A mechanically braked cycle ergometer (Monark Ergomedic 839E, Vansbro, Sweden) was utilized. Each patient chose his preferred cadence during practice trials, and this cadence was maintained during each repetition. 

During the first day the participants performed an incremental exercise test preceded by a 5 min resting measurement. The incremental exercise began with a 3-min warm-up at 30 W followed by 3 min at 60 W, then the work rate was increased by 20 W every minute until the participants reached voluntary exhaustion, defined as the inability to maintain the pedalling frequency despite vigorous encouragement by the researchers. For all variables, values determined at voluntary exhaustion during the incremental test were considered “peak” values. 

After 48 h without doing vigorous exercise, the patients performed one repetition of heavy-intensity constant-work rate (CWR) exercise to voluntary exhaustion. The intensity corresponded to 130% of the work rate at gas exchange threshold (GET), determined during the incremental exercise before the interventions. 

### 2.6. Measurements 

Pulmonary ventilation ($\dot{V}$_E_), tidal volume (V_T_), respiratory frequency (fR), O_2_ uptake ($\dot{V}$O_2_) and CO_2_ output ($\dot{V}$CO_2_) were determined on a breath-by-breath basis by means of a metabolic unit (MedGraphics CPX/D, Medical Graphics Corp., St Paul, MN, USA). Calibration of O_2_ and CO_2_ analyzers was performed before each experiment by utilizing gas mixtures of known composition. Expiratory flow measurements were performed by a bidirectional pressure differential pneumotachograph, which was calibrated by a 3-liter syringe at varying flow rates.

The respiratory gas-exchange ratio (R) was calculated as $\dot{V}$CO_2_/$\dot{V}$O_2_. Heart rate (HR) was determined by ECG. GET and the respiratory compensation threshold (RCT) were determined by the V-slope method; ventilatory equivalents ($\dot{V}$_E_/$\dot{V}$O_2_, $\dot{V}$_E_/$\dot{V}$CO_2_) and end-tidal partial pressures for O_2_ and CO_2_ (P_ET_O_2_ and P_ET_CO_2_) were utilized as ancillary signs [28]. Data related to GET and RCT are reported as $\dot{V}$O_2_ (L∙min^−1^) and corresponding work rate. Ratings of perceived exertion (RPE) for respiratory discomfort (RPE_R_) and limb effort (RPE_L_) were obtained at rest and every minute of exercise by using the Borg’s modified CR10 scale [29]. 

Considering that only one repetition of CWR could be carried out, and that the gas-exchange data are generally variable in young patients with obesity, formal $\dot{V}$O_2_ kinetics analyses were not performed [30]. Mean $\dot{V}$O_2_ values were calculated during the last three minutes of rest, before starting the exercise, and during the last 30 s of every minute of the CWR exercise. Individual linear regressions with a positive slope between the average values at the 3rd minute and at the last minute of exercise were indicative of the presence of a $\dot{V}$O_2_ slow component ($\dot{V}$O_2_ rise over time), and the difference between these values was used to determine its amplitude during the CWR exercises [18]. 

### 2.7. Statistical Analysis 

Results were expressed as means ± standard deviation (SD). 

Considering the primary outcome of this study, the change of the O_2_ demand during CWR cycling after RMIT, a sample size of 9 achieves 96% of power to detect a mean difference of 10% and a SD of differences of 7% with a significance level of 0.05. These percentages correspond to those previously observed in adolescents with obesity after RMET [18].

Significant differences between and within groups were checked by a two-way mixed ANOVA. A Bonferroni post hoc test was used to assess the significance of pair-wise comparisons. The level of significance was set at *p* < 0.05. Statistical analyses were carried out by utilizing commercially available software packages (Prism 9.2, GraphPad, San Diego, CA, USA; Statistical Package Social Sciences 15.0, IBM-SPSS Inc., Chicago, IL, USA). 

## 3. Results

### 3.1. Anthropometric and Pulmonary Outcomes

The main outcomes for anthropometry and pulmonary function are reported in Table 2. 

Both groups exhibited a significant decrease in BM (by ~4 kg) and BMI, which were associated with significant decreases in FFM in CTRL and FM in RMIT. 

No major signs of restrictive or obstructive alterations were observed in both groups before trainings. FVC, FEV_1_ and PEF improved significantly after RMIT, whereas no significant changes were observed after CTRL. 

A significant interaction between groups was observed for FVC (*p* = 0.02) and a nearly significant interaction was observed for FEV_1_ (*p* = 0.06). 

### 3.2. Cardiorespiratory Outcomes during the Incremental Exercise Test

The peak values of the main cardiorespiratory variables obtained during the incremental exercise are reported in Table 3. Before the interventions the two groups showed similar peak values in all the detected variables. At peak exercise the respiratory variables were not affected by the interventions, with the exception of peak V_T_ that increased significantly after RMIT (by 9.8 ± 9.0%). Peak $\dot{V}$O_2_ increased significantly after both interventions, either when expressed in absolute values (by 5.1 ± 3.7% after RMIT and 8.1 ± 6.7% after CTRL) or adjusted for BM (by 6.8 ± 4.3% after RMIT and 7.3 ± 6.7% after CTRL) and FFM (by 7.4 ± 4.9% after RMIT and 15.8 ± 6.9% after CTRL), whereas peak HR was significantly reduced (by 3.6 ± 3.0% after RMIT and 2.7 ± 2.6% after CTRL). As a result, peak WR increased significantly after both interventions (6.6 ± 4.6% after RMIT and 5.8 ± 4.1% after CTRL). GET values, peak RPE_R_ and RPE_L_ were not affected by the interventions. $\dot{V}$O_2_ at RCT increased significantly after CTRL but was not associated with improvements in the power output. A significant interaction between groups was observed for peak $\dot{V}$O_2_/FFM (*p* = 0.02).

### 3.3. Cardiorespiratory Outcomes during the Heavy-Intensity Exercise Test

The end-exercise values of the main cardiorespiratory variables obtained during the CWR exercise are reported in Table 4. The duration of the exercise increased in 8 out of 9 participants after RMIT (on average by 55.0 ± 24.7%), whereas it was not affected by the CTRL intervention. The interaction between groups was significant (*p* = 0.02). Notwithstanding a markedly greater duration of the exercise after RMIT, the end-exercise values of the cardiorespiratory responses and the perception of effort did not change. No significant changes were also observed after CTRL, in line with similar times to exhaustion. 

The mean (±SD) and the individual values of time to exhaustion during CWR cycling are shown in Figure 1 (RMIT) and Figure 2 (CTRL) together with the values of the main cardiorespiratory variables and perceived effort determined at the same time phase of the exercise (ISO-time) before and after the interventions. ISO-times correspond to the individual times to exhaustion reached before the intervention, or after the intervention in the case of decreased time to exhaustion (one participant of the RMIT group).

At ISO-time during CWR cycling, the values of $\dot{V}$O_2,_
$\dot{V}$_E_, and HR were significantly lower after RMIT (by 8.0 ± 4.8%, 10.0 ± 7.5%, 4.7 ± 2.6%, respectively) and were accompanied by a significantly lower perception of effort both for breathing and for the leg movements (see Figure 1), whereas only the perception of breath effort was significantly reduced after CTRL (see Figure 2). The interaction between groups was significant for $\dot{V}$O_2_ (*p* = 0.0009), and close to significance for $\dot{V}$_E_ (*p* = 0.05) and leg effort (*p* = 0.07). 

Figure 3a,b shows the typical patterns of $\dot{V}$O_2_ as a function of time during CWR>GET before and after RMIT and CTRL. In both groups, the exercise intensity elicited an additional increase in $\dot{V}$O_2_ after the first 3 min of exercise, denoting a slow component-like response. At the end of the exercise, the amplitudes of the $\dot{V}$O_2_ slow component did not differ after vs. before both interventions (see also Table 4).

The amplitudes of the increase in $\dot{V}$O_2_ from minute 3 to ISO-time were also calculated and shown in Figure 3c,d. At ISO-time, the $\dot{V}$O_2_ slow component was reduced in 8 out of 9 participants after RMIT (*p* = 0.0002), whereas it was not significantly affected by CTRL. The interaction between groups was significant (*p* = 0.0002).

Likewise, V_T_/T_I_ was similar after vs. before RMIT at minute 3 of the CWR exercise, but decreased significantly after RMIT at ISO-time (by 10.2 ± 7.1%, *p* = 0.02; Figure 4a); thus, the amplitude of the increase in V_T_/T_I_ during the CWR exercise (from minute 3 to ISO-time, ΔV_T_/T_I_) was also reduced after (0.34 ± 0.16 L∙s^−1^) vs. before (0.64 ± 0.20 L∙s^−1^) RMIT (*p* = 0.004). On the other hand, V_T_/T_I_ did not differ significantly after vs. before CTRL at any considered time phase of the CWR exercise (Figure 4b), thus ΔV_T_/T_I_ was also unchanged (0.38 ± 0.19 L∙s^−1^ after vs. 0.45 ± 0.15 L∙s^−1^ before CTRL; *p* = 0.97). The interaction between groups was significant for V_T_/T_I_ at ISO-time (*p* = 0.04) and for ΔV_T_/T_I_ (*p* = 0.03).

## 4. Discussion

### 4.1. Main Findings 

In a group of young patients with obesity, a supervised program of interval training targeted to the respiratory muscles, administered during a 3-week multidisciplinary BWRP (moderate caloric restriction, aerobic exercise training, psychological and nutritional counselling) decreased the cardiorespiratory burden, the overall O_2_ demand, and the perceived effort during constant work-rate cycling above the gas-exchange threshold. These changes were associated with a marked increase in the duration of exercise. By contrast, the same variables were not substantially affected by a sham protocol of respiratory training administered during the same BWRP. 

### 4.2. Characteristics of the Interval Training Approach for the Respiratory Muscles

This study evaluated the efficacy of a novel protocol of respiratory muscle training on the integrative response to exercise. This modality of respiratory muscle training was recently introduced and tested in normal-weight young adults by Spengler and colleagues, in comparison with the classical 30-min RMET protocol [20,21]. As observed by the authors, a sprint interval protocol characterized by short bouts of maximal hyperpnea with added resistance was able to induce similar levels of diaphragm and expiratory muscle fatigue than the RMET protocol, no clinically relevant adverse changes in the mechanical airway properties, and a significant increase in the respiratory muscle performance, with signs of increased activity of both the inspiratory and expiratory muscles after a 12-session program [21]. On the basis of these findings, we defined a 12-session program of respiratory muscle interval training adapted to young patients with obesity and evaluated its effects on cardiorespiratory and metabolic functions, and exercise capacity.

### 4.3. Metabolic Responses after Rmit and CTRL

The overall pulmonary $\dot{V}$O_2_ (metabolic cost of cycling) measured at ISO-time during CWR>GET was significantly lower (by 8.0%) after RMIT, whereas it was not affected by CTRL. Resting $\dot{V}$O_2_ values remain similarly high after RMIT and CTRL (Table 4), whereas in 8 out of 9 patients the reduced overall metabolic cost of cycling after RMIT was associated with a lower increase of $\dot{V}$O_2_ during the exercise, which denotes an attenuation in the development of the slow component (see Figure 3a–c). 

The $\dot{V}$O_2_ slow component normally heralds a decreased efficiency of muscle contractions intrinsic to the exercising locomotor muscles and, in a smaller part, to the respiratory muscles during prolonged heavy and severe-intensity exercises [31,32,33], yet it can be elicited at lower intensities, earlier time periods, and/or for greater portions in the presence of respiratory dysfunctions. Indeed, different previous studies observed changes in the amplitude of the $\dot{V}$O_2_ slow component once the mechanical loading of the respiratory muscles and/or the work of breathing was altered [34,35,36,37], demonstrating a strict relationship between these variables. 

In our previous study in adolescents with obesity [38], normoxic helium-O_2_ breathing attenuated the increase in $\dot{V}$O_2_ and the overall O_2_ cost during heavy-intensity cycling, suggesting that a substantial portion of the additional $\dot{V}$O_2_ in this population was associated with a bigger respiratory load. 

Of note, adolescents and young adults represent an age-range in which effective preventive/therapeutic interventions can break the detrimental loop obesity–breathlessness–inactivity which often causes the failure of any intervention in a medium-long time frame [39], thereby preventing the potential development of complications.

The results of the present study go in the same direction as those obtained after a 3-week program of respiratory muscle endurance training in a similar population [18]. In that study, a significant reduction in the $\dot{V}$O_2_ slow component and O_2_ cost of cycling was accompanied by a reduction in the abdominal ribcage hyperinflation and by a better recruitment of the lung and chest wall volumes [5]. 

### 4.4. Respiratory Responses and Exercise Tolerance after Rmit and CTRL

In the present study, the thoraco-abdominal operational pattern during exercise was not measured. Resting FVC, FEV_1_ and PEF were significantly increased after RMIT. During CWR>GET at ISO-time, the mean inspiratory flow and its increase during exercise were significantly reduced after RMIT, and were associated with a significant reduction in the ventilatory demand. The mean inspiratory flow is related to the inspiratory muscle work performed against the airway resistance [40], and its increase during CWR exercise is associated with the development of the $\dot{V}$O_2_ slow component [34,35,38]. 

Overall, the respiratory responses observed in this study after RMIT are signs of respiratory muscle unloading and improved respiratory function, and presumably elicit the cascade of metabolic improvements also observed after RMIT, i.e., the attenuation of the rise in $\dot{V}$O_2_ during the CWR exercise, the reduction in the metabolic cost of cycling, the lower perception of breathlessness and leg effort at the same time phase of exercise. The putative mechanism behind this link would reside in a bidirectional change of both respiratory and locomotor muscle blood flow with changes in the work of breathing, i.e., unloaded breathing would determine a decrease in the blood flow to the accessory respiratory muscles and an increase in the vascular conductance and blood flow to the exercising limbs, with associated reduction in limb fatigue and improvement of the exercise performance [41,42].

Indeed, exercise tolerance was markedly improved after RMIT in 8 out of 9 patients (Δ increase: 4.2 ± 1.9 min). On the other hand, time to exhaustion was not significantly altered by CTRL, in association with the absence of change in the cardiorespiratory variables and leg effort. CTRL induced lower breath effort during CWR>GET, which could be associated with a general increase in the fitness level of the patients induced by the program of whole-body exercise, as indeed demonstrated by the significant increase in $\dot{V}$O_2_peak and $\dot{V}$O_2_ at RCT. 

### 4.5. Limitations and Perspectives

A limitation of the present study is that the two interventions could not be randomized within the same group of patients and, although the main characteristics of the two groups at baseline were similar, the CTRL group showed a better, not significant, exercise performance during the CWR exercise before the intervention. The reasons for this discrepancy could be related to inter- and intra-individual variations in the responses to exercise, observed in young patients with obesity, and/or to differences in the exercise intensity relative to RCT, and/or to small underestimations of the actual load during the CWR exercise. Baseline characteristics, however, have been taken into consideration in the statistical analysis.

Moreover, the specific effects of the RMIT protocol cannot be disentangled from the general effects of the multidisciplinary intervention and we recognize that the individual responses may be the result of different adaptations, more or less manifest, at different levels of the organism induced by the variety of stimuli and their combination. However, we believe that the consistency in the direction of the responses for all the main detected variables after RMIT and not after CTRL strongly supports RMIT as a possible intervention for improving exercise tolerance in this population. 

Moreover, in this study we did not examine the micronutrients of the diet nor muscle strength; both assessments could be pursued in future investigations.

Finally, the interval protocol applied in this study was defined on the basis of the responsiveness of a small pilot group of patients, by adopting a “safe” approach which limited the effort and was well tolerated by the participants (see Table 1). We believe that this training regimen has large margins to be changed, improved and tailored to individual needs; therefore, further investigations in this direction and on the mechanisms behind its systemic effects would be useful. In a future perspective, we believe that this approach could be strategically combined with motor control exercises directed to the lumbar-pelvic region, and particularly to the diaphragm, transverse abdomen and pelvic floor [13], in order to enhance and stabilize the physical adaptations that concur to improve the respiratory function, the perception of breathlessness and exercise tolerance, as well as to increase the patients’ awareness for their body, posture and breathing modality during their daily life.

## 5. Conclusions 

In a group of young patients with obesity, a 3-week supervised program of interval training for the respiratory muscles (normocapnic hyperpnea with inspiratory and expiratory resistances), administered during a multidisciplinary BWRP, decreased the cardiorespiratory burden, the overall metabolic cost of exercise and the perceived effort during constant work-rate cycling above the gas-exchange threshold, and globally was well tolerated. These responses were associated with a marked increase in the duration of the exercise. 

These findings demonstrate the need and feasibility to include specific programs of respiratory muscle training within the current interventions of body mass reduction and health care of patients with obesity, both in hospital and at home. Indeed, the respiratory muscle training may represent a key strategy to help patients with obesity, and other patients affected by cardiopulmonary dysfunctions and/or profound muscle weakness, to overcome the perception of breathlessness and the respiratory limitations that frequently induce them to interrupt/reduce/avoid whole-body exercise programs. 

## Figures and Tables

**Figure 1 ijerph-20-00487-f001:**
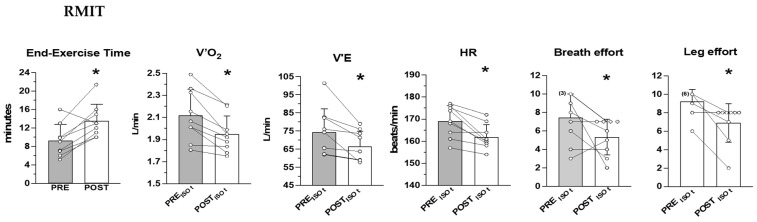
RMIT. Mean (±SD) and individual values of time to exhaustion, and ISO-time $\dot{V}$O_2_, $\dot{V}$_E_, HR, perceived effort for breathing (RPE_R_) and leg movements (RPE_L_) determined during constant work rate cycling at 130% of the work rate at GET, before (PRE) and after (POST) the RMIT program. ISO-time corresponds to the individual time to exhaustion before the intervention, or after the intervention for the participant who reduced the duration of the exercise. The number of overlapped data is reported in parenthesis next to the symbol. * significant difference vs. PRE (Time to exhaustion, *p* = 0.0003; $\dot{V}$O_2,_
*p* = 0.0007; $\dot{V}$_E_, *p* = 0.01; HR, *p* = 0.02; RPE_R_, *p* = 0.03; RPE_L_, *p* = 0.0003).

**Figure 2 ijerph-20-00487-f002:**
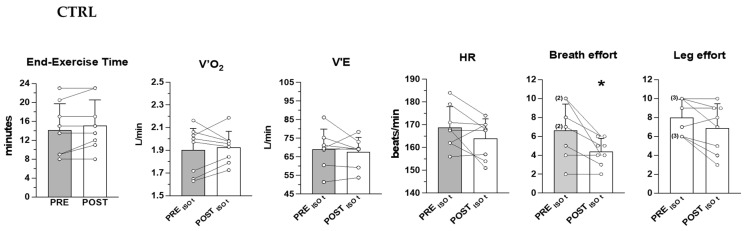
CTRL. Mean (±SD) and individual values of time to exhaustion, and ISO-time $\dot{V}$O_2,_
$\dot{V}$_E_, HR, perceived effort for breathing (RPE_R_) and leg movements (RPE_L_) determined during constant work rate cycling at 130% of the work rate at GET, before (PRE) and after (POST) the CTRL program. ISO-time corresponds to the individual time to exhaustion before the intervention. The number of overlapped data is reported in parenthesis next to the symbol. * significant difference vs. PRE (RPE_R_, *p* = 0.03).

**Figure 3 ijerph-20-00487-f003:**
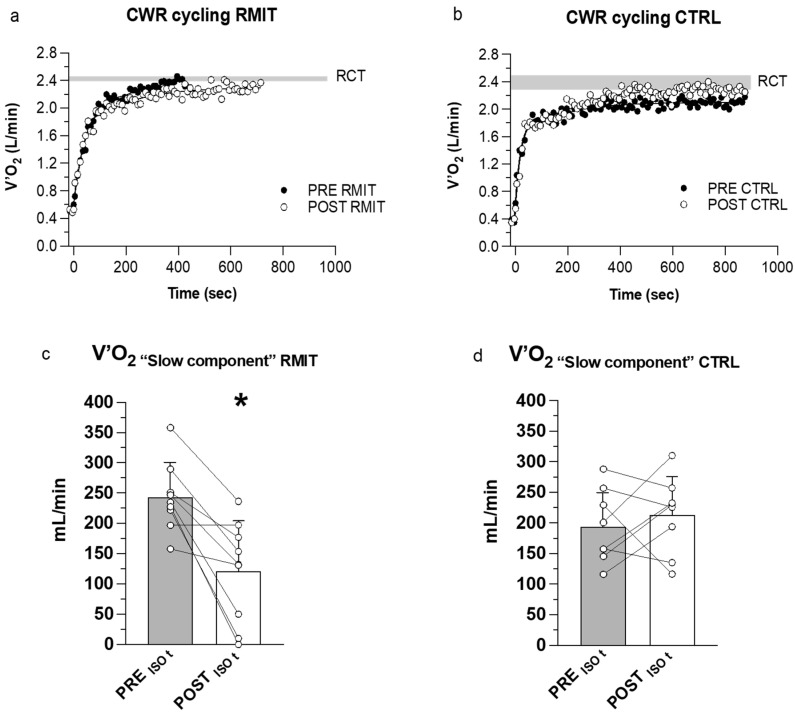
(**a**,**b**) Representative patterns of $\dot{V}$O_2_ as a function of time during CWR>GET carried out before and after RMIT and CTRL; (**c**,**d**) Mean (±SD) and individual values of the absolute increase in $\dot{V}$O_2_ from the third minute of exercise to ISO-time, for RMIT and CTRL. * significant difference vs. PRE (*p* = 0.0002).

**Figure 4 ijerph-20-00487-f004:**
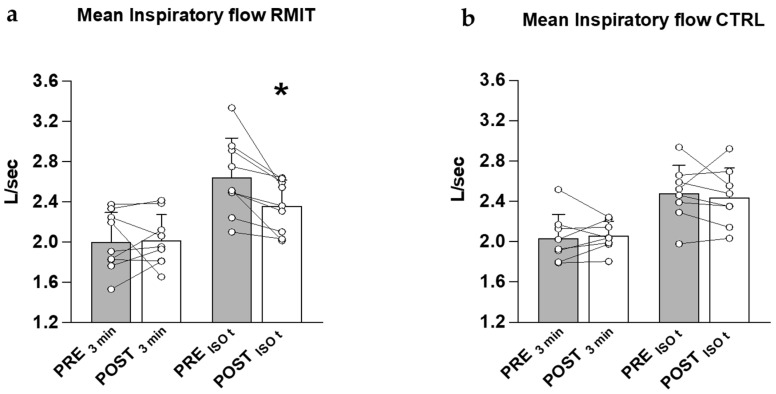
(**a**,**b**). Mean (±SD) and individual values of V_T_/T_I_ at the third minute and ISO-time of CWR>GET, for RMIT and CTRL. * significant difference vs. PRE (*p* = 0.02).

**Table 1 ijerph-20-00487-t001:** Physiological responses during the CTRL and RMIT protocols.

**CTRL (*n* = 8)**							
**Session 1**							
	**Before**	**End I1**	**End I2**	**End I3**	**End I4**	**End I5**	**End I6**
HR bpm	89.5 ± 8.4	89.9 ± 8.5	91.1 ± 8.6	91.4 ± 8.8	91.9 ± 8.2	91.6 ± 7.6	91.5 ± 6.8
RPE_B_	0.0 ± 0.0	0.0 ± 0.0	0.0 ± 0.0	0.0 ± 0.0	0.0 ± 0.0	0.0 ± 0.0	0.0 ± 0.0
SpO_2_ %	95.9 ± 0.6	-	-	-	-	-	96.1 ± 0.6
SBP mmHg	115.6 ± 6.2	-	-	-	-	-	118.1 ± 3.7
DBP mmHg	81.2 ± 8.3	-	-	-	-	-	81.2 ± 6.4
**Session 12**							
	**Before**	**End I1**	**End I2**	**End I3**	**End I4**	**End I5**	**End I6**
HR bpm	87.7 ± 9.3	90.0 ± 7.9	90.4 ± 8.0	91.7 ± 9.2	90.6 ± 7.8	90.3 ± 7.6	90.3 ± 8.2
RPE_B_	0.0 ± 0.0	0.0 ± 0.0	0.0 ± 0.0	0.0 ± 0.0	0.0 ± 0.0	0.0 ± 0.0	0.0 ± 0.0
SpO_2_ %	96.1 ± 0.6	-	-	-	-	-	96.4 ± 0.5
SBP mmHg	115.6 ± 4.9	-	-	-	-	-	118.7 ± 3.5
DBP mmHg	80.0 ± 2.7	-	-	-	-	-	82.5 ± 4.6
**RMIT (*n* = 9)**							
**Session 1**							
	**Before**	**End I1**	**End I2**	**End I3**	**End I4**	**End I5**	**End I6**
HR bpm	89.9 ± 9.2	96.8 ± 9.5	105.8 ± 8.1	107.0 ± 10.4	106.4 ± 7.2	107.5 ± 8.5	105.6 ± 9.1
RPE_B_	0.0 ± 0.0	4.0 ± 1.9	4.0 ± 1.4	4.4 ± 1.7	5.2 ± 2.4	4.8 ± 1.9	4.8 ± 1.9
SpO_2_ %	95.0 ± 0.5	-	-	-	-	-	95.4 ± 0.8
SBP mmHg	120.6 ± 6.3	-	-	-	-	-	120.4 ± 3.2
DBP mmHg	85.0 ± 8.3	-	-	-	-	-	86.6 ± 8.2
**Session 12**							
	**Before**	**End I1**	**End I2**	**End I3**	**End I4**	**End I5**	**End I6**
HR bpm	91.6 ± 3.7	98.2 ± 8.0	104.0 ± 9.9	104.6 ± 9.8	106.9 ± 9.9	108.7 ± 8.7	104.9 ± 9.8
RPE_B_	0.0 ± 0.0	0.5 ± 0.9	0.7 ± 1.3	0.7 ± 1.3	0.8 ± 1.3	1.1 ± 1.7	1.3 ± 2.1
SpO_2_ %	95.4 ± 1.0	-	-	-	-	-	95.8 ± 1.1
SBP mmHg	120.0 ± 3.9	-	-	-	-	-	120.8 ± 4.9
DBP mmHg	81.6 ± 4.1	-	-	-	-	-	81.9 ± 4.5

Means ± standard deviations of data obtained before the CTRL and RMIT protocols and at the immediate end of each 2-min interval (End I), during the first and the last session of both trainings. CTRL, control comparison. RMIT, respiratory muscle interval training intervention. *n*, no. of patients. HR, heart rate. RPE_B_, rate of perceived exertion for respiratory discomfort. SpO_2_, arterial blood O_2_ saturation. SBP, systolic blood pressure. DBP, diastolic blood pressure.

**Table 2 ijerph-20-00487-t002:** Anthropometric characteristics and pulmonary function of the patients before and after the standard intervention of body mass reduction combined with control training and with RMIT.

	CTRL (*n* = 8)			RMIT (*n* = 9)		
	Before	After	*p*	Before	After	*p*
Age (years)	17.0 ± 5.7	17.0 ± 5.7	1.00	17.9 ± 4.9	17.9 ± 4.9	1.00
Stature (m)	1.72 ± 0.06	1.72 ± 0.06	1.00	1.72 ± 0.08	1.72 ± 0.08	1.00
Body mass (kg)	109.3 ± 8.9	105.1 ± 9.1	**0.0002**	113.8 ± 16.3	110.0 ± 16.2	**0.0001**
BMI (kg∙m^−2^)	37.0 ± 3.2	35.6 ± 3.8	**0.0003**	38.6 ± 5.5	37.3 ± 5.3	**0.0001**
FFM (kg)	65.8 ± 6.9	62.4 ± 6.1	**0.009**	69.8 ± 12.9	68.2 ± 12.7	0.11
FM (kg)	43.5 ± 3.3	42.7 ± 4.2	0.23	44.0 ± 3.9	41.9 ± 5.1	**0.02**
FM (% of body mass)	40.0 ± 1.3	40.3 ± 1.0	0.99	39.0 ± 2.7	38.3 ± 3.5	0.71
FVC (L)	4.82 ± 0.25	4.80 ± 0.35	0.74	4.55 ± 0.63	5.01 ± 0.50	**0.01**
FVC (% of predicted)	102.2 ± 5.3	101.8 ± 7.4	0.79	93.8 ± 13.0	103.3 ± 10.3	**0.006**
FEV_1_ (L)	4.05 ± 0.32	4.00 ± 0.31	0.80	3.80 ± 0.37	4.22 ± 0.55	**0.02**
FEV_1_ (% of predicted)	100.7 ± 7.9	99.5 ± 7.7	0.88	92.1 ± 8.9	102.3 ± 12.3	**0.02**
FEV_1_/FVC (%)	83.9 ± 5.2	82.8 ± 4.4	0.93	84.1 ± 6.7	83.7 ± 6.7	0.86
FEF_25–75%_ (L∙min^−1^)	4.35 ± 0.79	4.25 ± 0.58	0.58	3.99 ± 0.62	4.57 ± 0.99	0.12
FEF_25–75%_ (% of predicted)	100.1 ± 18.2	97.8 ± 13.3	0.66	89.3 ± 13.9	102.2 ± 22.1	0.10
PEF (L∙s^−1^)	7.39 ± 1.49	7.57 ± 1.36	0.67	6.64 ± 0.79	8.10 ± 1.40	**0.01**
PEF (% of predicted)	87.1 ± 19.6	89.2 ± 16.0	0.68	78.4 ± 11.9	97.6 ± 18.8	**0.004**

Data are given as means ± standard deviations. CTRL, control comparison. RMIT, respiratory muscle interval training intervention. *n*, no. of patients. BMI, body mass index; FFM, fat-free mass; FM, fat mass. FVC, forced vital capacity; FEV_1_, forced expiratory volume in 1 s; FEF25–75%, forced expiratory flow between 25% and 75% of FVC; PEF, peak expiratory flow. *p*-values obtained from 2-way mixed ANOVA and Bonferroni post hoc correction.

**Table 3 ijerph-20-00487-t003:** Peak values of the main investigated variables, determined at exhaustion during the incremental exercise, before and after the standard intervention of body mass reduction combined with a control training and with RMIT.

	CTRL (*n* = 8)			RMIT (*n* = 9)		
	Before	After	*p*	Before	After	*p*
Work rate (watt)	148.7 ± 13.6	157.5 ± 17.5	**0.005**	154.4 ± 21.3	164.4 ± 22.4	**0.001**
HR (b∙min^−1^)	173.7 ± 6.9	169.1 ± 6.8	**0.04**	179.3 ± 3.6	172.8 ± 6.6	**0.003**
$\dot{V}$_E_ (L∙min^−1^)	77.4 ± 9.2	84.8 ± 11.7	0.25	80.4 ± 23.9	84.3 ± 13.3	0.62
V_T_ (L)	2.02 ± 0.46	2.10 ± 0.46	0.39	2.05 ± 0.38	2.25 ± 0.56	**0.01**
fR (br∙min^−1^)	39.4 ± 6.6	41.3 ± 5.9	0.78	39.4 ± 9.6	38.6 ± 6.1	0.96
$\dot{V}$O_2_ (L∙min^−1^)	2.22 ± 0.34	2.40 ± 0.34	**0.0007**	2.34 ± 0.32	2.46 ± 0.30	**0.02**
$\dot{V}$O_2_/BM (mL∙kg^−1^∙min^−1^)	20.5 ± 4.0	22.0 ± 3.9	**0.0007**	20.5 ± 2.3	21.9 ± 2.5	**0.02**
$\dot{V}$O_2_/FFM (mL∙kg^−1^∙min^−1^)	33.5 ± 8.1	38.8 ± 7.8	**0.0001**	33.8 ± 3.9	36.3 ± 5.2	**0.003**
$\dot{V}$CO_2_ (L∙min^−1^)	2.40 ± 0.31	2.52 ± 0.31	0.07	2.48 ± 0.42	2.65 ± 0.32	0.14
R	1.08 ± 0.10	1.05 ± 0.08	0.42	1.05 ± 0.10	1.08 ± 0.07	0.50
$\dot{V}$_E_/$\dot{V}$O_2_	35.2 ± 6.6	35.6 ± 5.0	0.99	33.4 ± 10.0	34.8 ± 5.8	0.85
$\dot{V}$_E_/$\dot{V}$CO_2_	31.9 ± 3.1	33.7 ± 2.1	0.36	31.2 ± 6.1	32.1 ± 3.5	0.94
PETO_2_ (mmHg)	95.6 ± 4.5	96.1 ± 3.4	0.99	92.8 ± 7.4	95.0 ± 4.3	0.32
PETCO_2_ (mmHg)	34.7 ± 3.2	33.4 ± 2.0	0.55	36.4 ± 5.1	34.8 ± 3.4	0.35
RPE_R_	7.5 ± 1.8	7.0 ± 1.2	0.99	8.0 ± 1.7	6.9 ± 2.6	0.36
RPE_L_	8.9 ± 1.1	9.1 ± 1.0	0.90	9.3 ± 0.9	9.4 ± 0.5	0.99
GET (L∙min^−1^)	1.33 ± 0.28	1.43 ± 0.23	0.25	1.57 ± 0.24	1.50 ± 0.21	0.95
GET (watt)	87.5 ± 11.6	87.5 ± 11.6	1.00	96.3 ± 7.4	90.0 ± 15.0	0.41
RCT (L∙min^−1^)	2.15 ± 0.42	2.35 ± 0.29	**0.01**	2.29 ± 0.40	2.41 ± 0.35	0.59
RCT (watt)	143.7 ± 16.0	152.5 ± 18.3	0.12	146.3 ± 26.1	158.6 ± 25.5	0.20

Data are given as means ± standard deviations. *n*, no. of patients. CTRL, control comparison. RMIT, respiratory muscle interval training intervention. HR, heart rate; $\dot{V}$_E_, pulmonary ventilation, V_T_, tidal volume; fR, respiratory frequency; $\dot{V}$O_2_, O_2_ uptake; BM, body mass; FFM: fat-free mass; $\dot{V}$CO_2_, CO_2_ output; R, respiratory gas-exchange ratio, PETO_2_, O_2_ end-tidal pressure; PETCO_2_, CO_2_ end-tidal pressure; RPE_R_, rate of perceived exertion for respiratory discomfort; RPE_L_, rate of perceived exertion for leg effort; GET, gas exchange threshold; RCT, respiratory compensation threshold. *p*-values obtained from 2-way mixed ANOVA and Bonferroni post hoc correction.

**Table 4 ijerph-20-00487-t004:** End-exercise values of the main investigated variables determined during the last minute of the constant work rate exercise carried out at 130% of the work rate at GET, before and after the standard intervention of body mass reduction combined with a control training and with RMIT.

	CTRL (*n* = 8)			RMIT (*n* = 9)		
	Before	After	*p*	Before	After	*p*
Work rate (watt)	113.7 ± 15.1	113.7 ± 15.1	1.00	122.3 ± 12.3	122.3 ± 12.3	1.00
Time to exhaustion (min)	14.4 ± 5.6	15.3 ± 5.4	0.62	9.2 ± 3.5 *	13.5 ± 3.7	**0.0003**
HR (b∙min^−1^)	168.7 ± 9.2	164.4 ± 8.3	0.30	169.3 ± 6.9	169.5 ± 5.9	0.99
$\dot{V}$_E_ (L∙min^−1^)	68.0 ± 11.1	69.5 ± 7.7	0.88	73.9 ± 13.7	73.1 ± 11.7	0.80
V_T_ (L)	1.72 ± 0.36	1.80 ± 0.34	0.31	1.98 ± 0.28	1.96 ± 0.49	0.99
fR (br min^−1^)	40.6 ± 9.3	39.6 ± 7.7	0.72	37.7 ± 7.2	38.5 ± 6.6	0.98
V_T_/T_I_ (L∙s^−1^)	2.48 ± 0.28	2.54 ± 0.36	0.98	2.64 ± 0.38	2.56 ± 0.40	0.76
Resting $\dot{V}$O_2_ (L∙min^−1^)	0.42 ± 0.05	0.44 ± 0.03	0.90	0.44 ± 0.08	0.44 ± 0.07	0.99
$\dot{V}$O_2_ (L∙min^−1^)	1.87 ± 0.20	1.96 ± 0.11	0.27	2.11 ± 0.25	2.08 ± 0.17	0.90
$\dot{V}$O_2_ slow component (L∙min^−1^)	0.19 ± 0.06	0.25 ± 0.08	0.51	0.24 ± 0.05	0.25 ± 0.15	0.99
$\dot{V}$O_2_/BM (mL∙kg^−1^∙min^−1^)	17.1 ± 2.9	18.0 ± 2.8	0.21	18.6 ± 2.15	18.5 ± 1.2	0.96
$\dot{V}$O_2_/FFM (mL∙kg^−1^∙min^−1^)	28.3 ± 5.9	31.7 ± 5.2	**0.03**	30.8 ± 4.6	31.2 ± 3.7	0.88
$\dot{V}$CO_2_ (L∙min^−1^)	1.86 ± 0.19	1.88 ± 0.14	0.94	2.10 ± 0.23	2.06 ± 0.21	0.44
R	1.00 ± 0.07	0.97 ± 0.05	0.62	1.00 ± 0.10	0.99 ± 0.04	0.70
$\dot{V}$_E_/$\dot{V}$O_2_	33.8 ± 4.9	32.9 ± 4.0	0.59	32.5 ± 7.8	32.5 ± 4.3	0.99
$\dot{V}$_E_/$\dot{V}$CO_2_	33.7 ± 3.4	34.1 ± 2.1	0.84	32.1 ± 3.9	32.9 ± 3.0	0.64
PETO_2_ (mmHg)	95.1 ± 4.5	94.8 ± 3.5	0.92	92.9 ± 4.7	94.0 ± 3.4	0.62
PETCO_2_ (mmHg)	33.1 ± 3.6	32.2 ± 2.0	0.53	34.9 ± 3.1	33.6 ± 2.6	0.15
RPE_R_	6.6 ± 2.8	4.9 ± 1.2	0.10	7.8 ± 2.8	7.7 ± 2.5	0.99
RPE_L_	8.0 ± 1.9	7.6 ± 2.8	0.99	9.0 ± 1.7	9.6 ± 0.9	0.99

Data are given as means ± standard deviations. *n*, no. of patients. CTRL, control comparison. RMIT, respiratory muscle interval training intervention. HR, heart rate; $\dot{V}$_E_, pulmonary ventilation, V_T_, tidal volume; fR, respiratory frequency; V_T_/T_I_, mean inspiratory flow rate; $\dot{V}$O_2_, O_2_ uptake; RCT, respiratory compensation threshold; GET, gas exchange threshold; BM, body mass; FFM: fat-free mass; $\dot{V}$CO_2,_ CO_2_ output; R, respiratory gas-exchange ratio, PETO_2_, O_2_ end-tidal pressure; PETCO_2_, CO_2_ end-tidal pressure; RPE_R_, rate of perceived exertion for respiratory discomfort; RPE_L_, rate of perceived exertion for leg effort. *p-*values obtained from 2-way mixed ANOVA and Bonferroni post-hoc correction. * *p* = 0.06 between groups at baseline.

## Data Availability

Raw data will be available upon reasonable request to the corresponding author and will be uploaded on zenodo.org immediately after the acceptance of the manuscript.

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
