# Peer review of "Respiratory Muscle Interval Training Improves Exercise Capacity in Obese Adolescents during a 3-Week In-Hospital Multidisciplinary Body Weight Reduction Program"

_ijerph, 2022, doi:10.3390/ijerph20010487_

Round 1
Reviewer 1 Report
It is with sincere pleasure that I look over the study entitled " Respiratory Muscle Interval Training: a novel approach to improve exercise capacity in young people with obesity".
This study has a quite high degree of utility for physicians and exercise physiologists to combat obesity. Please see my comments below to improve the quality of the manuscript.
First, I would permanently have a preference to review the methods section preceding reading over other sections.
Please describe how you evaluated body composition. Along with CVs and ICCs, I would appreciate it if you could specify the devices utilized. If you assessed body composition using BIA impedance, do you have skeletal muscle mass data? It is far superior to FFM.
Please report at least all macronutrients since the participants were under dietitian supervision, and no nutrients were assessed in this research. Unfortunately, it appears that the majority of clinical nutritionists do not prioritize protein intake during physical activity. Despite the fact that research indicates that obesity alters muscle protein synthesis responses to nutrition and physical activity, particularly resistance training (PMID: 31263701), it is highly likely that increased protein intake reduces muscle loss during weight loss interventions.
In addition, the evaluation of muscle strength may be a valuable indicator of muscular growth during exercise. Do you have any data on this?
The writing quality seems good throughout the introduction and the rest of the sections.
As soon as I receive the author's answer to my suggestions, I intend to give the other sections my entire attention.
Author Response
It is with sincere pleasure that I look over the study entitled " Respiratory Muscle Interval Training: a novel approach to improve exercise capacity in young people with obesity".
"This study has a quite high degree of utility for physicians and exercise physiologists to combat obesity."
We would like to thank Reviewer 1 for her/his appreciation and for the thoughtful and constructive review of our manuscript. Below you can see a point-by-point answer to your queries. We are confident that the new version of the manuscript fulfills all the requirements and changes requested. These modifications have been written using “Track Changes”.
Q1. Please describe how you evaluated body composition. Along with CVs and ICCs, I would appreciate it if you could specify the devices utilized. If you assessed body composition using BIA impedance, do you have skeletal muscle mass data? It is far superior to FFM.
A1. The lacking information regarding CVs and ICCs have been added. Information regarding the device were already reported in the text (line 90, p. 2). Taking into account the extremely high BMI of our study population (which could influence the skeletal muscle mass evaluation by BIA), we opted to consider only FM and FFM.
Q2. Please report at least all macronutrients since the participants were under dietitian supervision, and no nutrients were assessed in this research. Unfortunately, it appears that the majority of clinical nutritionists do not prioritize protein intake during physical activity. Despite the fact that research indicates that obesity alters muscle protein synthesis responses to nutrition and physical activity, particularly resistance training (PMID: 31263701), it is highly likely that increased protein intake reduces muscle loss during weight loss interventions.
A2. The study group was hospitalized for a 3-week multidisciplinary body weight reduction program, entailing an energy-restricted diet, physical rehabilitation (moderate aerobic activity), psychological counseling and nutritional education. One of the aim of our short-term program of metabolic rehabilitation is to progressively re-educate the young patients to a healthy life-style (as far as nutrition and exercise are concerned) to be maintained at home. The impact of a markedly increased protein intake might be accepted with difficulty in subjects usually accustomed to a diet rich in carbohydrates, thus increasing the risk of a poor compliance. Information regarding macronutrients has been added.
Q3. In addition, the evaluation of muscle strength may be a valuable indicator of muscular growth during exercise. Do you have any data on this?
A3. The present study was focused to study the effects of a specific protocol of respiratory muscle training on the cardiorespiratory burden, the O2 demand and the perception of effort during heavy-intensity cycling and, ultimately, on the exercise capacity of obese young patients. No evaluation of muscle strength has been performed. The evaluation of muscle strength will be possibly taken into consideration in future studies.
Q4. The writing quality seems good throughout the introduction and the rest of the sections.
A4. Thank you for the appreciation of our work.
Reviewer 2 Report
Dear authors,
The study is interesting, with many variables assessed in obesity young patients under a multidisciplinary program with or not respiratory muscle training. The authors show many results sustained in the study design; however, it is difficult to separate or infer that the outcomes achieved are results of the respiratory muscles training. This topic is complex because the title and the introduction should be changed to report the results coherently with the main ideas exposed in the backgrounds. The authors compared many variables by t-test and two-way ANOVA, but there was no “common line” in the results. So, in my opinion, the study should be rewriter, and the authors to be guided by the PICOT system (“P” participants or patients; “I” intervention; “C” comparison or control; “o” main outcomes; and “T” time or type of study). My suggestion is to reject the article as today it is written.
Author Response
"The study is interesting, with many variables assessed in obesity young patients under a multidisciplinary program with or not respiratory muscle training. The authors show many results sustained in the study design; however, it is difficult to separate or infer that the outcomes achieved are results of the respiratory muscles training."
We would like to thank Reviewer 2 for her/his appreciation and for the thoughtful and constructive review of our manuscript. Below you can see a point-by-point answer to your queries. We are confident that the new version of the manuscript fulfills all the requirements and changes requested. These modifications have been written using “Track Changes”.
Q1. This topic is complex because the title and the introduction should be changed to report the results coherently with the main ideas exposed in the backgrounds.
A1. As suggested, the title and introduction have been changed.
Q2. The authors compared many variables by t-test and two-way ANOVA, but there was no “common line” in the results. So, in my opinion, the study should be rewriter, and the authors to be guided by the PICOT system (“P” participants or patients; “I” intervention; “C” comparison or control; “o” main outcomes; and “T” time or type of study). My suggestion is to reject the article as today it is written.
A2. As requested, the results section has been modified according with the PICOT system.
Reviewer 3 Report
1. In table 1 Physiological responses during CTRL and RMIT protocols, CTRL is omitted from the description.
2. It is interesting to introduce a flow chart of the study.
Author Response
We would like to thank Reviewer 3 for her/his time and effort for the review of our work. Below you can see a point-by-point answer to your queries. We are confident that the new version of the manuscript fulfills all the requirements and changes requested. These modifications have been written using “Track Changes”.
Q1. In table 1 Physiological responses during CTRL and RMIT protocols, CTRL is omitted from the description.
A1. Done
Q2. It is interesting to introduce a flow chart of the study.
A2. We believe that a flow chart could weigh down the manuscript, which already contains 4 figures and 4 tables.
Round 2
Reviewer 1 Report
Thank you for addressing my comments.
Please report the exact model of BIA. Single frequency or multi? if single, which model? If multi, which model? This is critical.
Please include the lack of nutrient data in the limitations.
Also, include assessing muscle strength for future studies.
Author Response
Q1. Please report the exact model of BIA. Single frequency or multi? if single, which model? If multi, which model? This is critical.
A1. Fat-free mass (FFM) and fat mass (FM) were determined by a multifrequency impedance meter (Human IM touch®, DS-Medica, Milan, Italy). This is now specified in the revised text (Line 97, p. 2).
Q2. Please include the lack of nutrient data in the limitations.
A2. A sentence that recognizes the limitation related to the lack of information regarding some micronutrients has now been added in the revised text (Lines 460-461 p. 14).
Q3. Also, include assessing muscle strength for future studies.
A3. A sentence has been added in the revised text (Lines 460-461 p. 14).
Reviewer 2 Report
Dear authors,
Thanks for improving the manuscript; although your are incorporated many interesting ideas in this version, the study design could not allow us to infer that the changes shown in the primary outcome have been attributed to respiratory muscle training, as your support in the manuscript. In my opinion, the manuscript should be rewritten.
Author Response
Q1. Thanks for improving the manuscript; although you are incorporated many interesting ideas in this version, the study design could not allow us to infer that the changes shown in the primary outcome have been attributed to respiratory muscle training, as your support in the manuscript. In my opinion, the manuscript should be rewritten.
A1. Since the patients are normally hospitalized for a 3-week BWRP which is planned and reimbursed by our NHS we are obliged to prescribe a package of activities including energy restricted diet and nutritional education, physical activity, and psychological counselling. Thus, we cannot design and evaluate the effects of RMIT alone. Anyway, in the present study we also evaluated a randomly chosen BMI- and age-matched CONTROL group (performing a sham protocol of respiratory training), following the same BWRP of the RMIT subgroup in term of physical activity, nutritional intake and psychological counselling. The Reviewer is right, we cannot disentangle the specific effects of the RMIT protocol from the general effects of the multidisciplinary intervention, this was already recognized in the manuscript (Lines 453-456, p. 13-14), but we believe that the consistency in the direction of the responses for all the main detected variables after RMIT and not after CTRL strongly supports in favor of RMIT as a possible intervention for improving exercise tolerance in this population.